# It's My Data Too: Private ML for Datasets with Multi-User Training Examples

Arun Ganesh [1]   Ryan McKenna [1]   Brendan McMahan [1]   Adam Smith [2]   Fan Wu [3]

## Abstract

We initiate a study of algorithms for model training with user-level differential privacy (DP), where each example may be attributed to multiple users, which we call the multi-attribution model. We first provide a carefully chosen definition of user-level DP under the multi-attribution model. Training in the multi-attribution model is facilitated by solving the contribution bounding problem, i.e. the problem of selecting a subset of the dataset for which each user is associated with a limited number of examples. We propose a greedy baseline algorithm for the contribution bounding problem. We then empirically study this algorithm for a synthetic logistic regression task and a transformer training task, including studying variants of this baseline algorithm that optimize the subset chosen using different techniques and criteria. We find that the baseline algorithm remains competitive with its variants in most settings, and build a better understanding of the practical importance of a bias-variance tradeoff inherent in solutions to the contribution bounding problem.

## 1. Introduction

We study private model training with user-level differential privacy (DP) (Dwork et al., 2010), where each example is associated with multiple users (and in particular, might contain privacy-sensitive information for any or all associated users). This goes beyond previously-studied settings for user-level DP which (implicitly) adopt the *single-attribution model*, where each example is attributed to a single user. This assumption greatly simplifies the problem, as it allows one to reduce to algorithms tailored to example-level DP, by e.g. computing the average gradient for each user's exam-

ples and then using it as an example gradient in an algorithm like DP-SGD (this is exactly the DP-FedSGD algorithm introduced in (McMahan et al., 2017)). However, without additional assumptions this approach only provides provable protections when an example's attributee is the sole person whose privacy is affected by leakage of that example; this is often untrue in practice. For example, a SMS or email message may contain privacy-sensitive information about both the sender and receivers, but might only be attributed to the sender in the single-attribution model.

Motivated by this gap, we propose and study the stronger and more flexible *multi-attribution model*: in addition to multiple examples being attributed to the same user, each example is potentially attributed to multiple users. This strictly generalizes the single-attribution model and extends to other practical settings not previously covered, but comes with a number of algorithmic difficulties. First, even defining DP in the multi-attribution model requires extra care. Even with an appropriate definition, user-to-example reductions for algorithms like DP-FedSGD can't be directly applied to multi-attribution data due to the overlap in users' data, and more nuanced data selection algorithms are required.

Figure 1 compares the multi-attribution setting we consider to other more standard settings, where training examples are formed from emails sent among different users (perhaps to train a language model or spam classifier).

### 1.1. Our Contributions

**Defining DP with multi-attribution:** In Section 2, we propose a definition for DP with multiply attributed data, which we dub *fixed-graph DP*. In this model, we have a hypergraph where vertices represent users, and each hyperedge is associated with an example attributed to its vertices. Our definition protects against changing the content of examples associated with hyperedges adjacent to a single vertex. More precisely, our adjacency definition considers two datasets adjacent if they have the same hypergraph, but differ in (potentially) all of the data associated with a single vertex. We compare our approach to several alternatives, including a generalization of *node privacy* to hypergraphs (Fang et al., 2022). Continuing with the example of email data, our approach allows a data curator to make statements to users like: *"Even if every email you ever sent **or received**"*

*Equal contribution [1]Google Research, Seattle, USA [2]Boston University and Google DeepMind, Cambridge, USA [3]University of Illinois, Champaign–Urbana, USA. Part of this work done while the author was an intern at Google Research.. Correspondence to: Arun Ganesh <arunganesh@google.com>.

*Proceedings of the 42$^{nd}$ International Conference on Machine Learning*, Vancouver, Canada. PMLR 267, 2025. Copyright 2025 by the author(s).

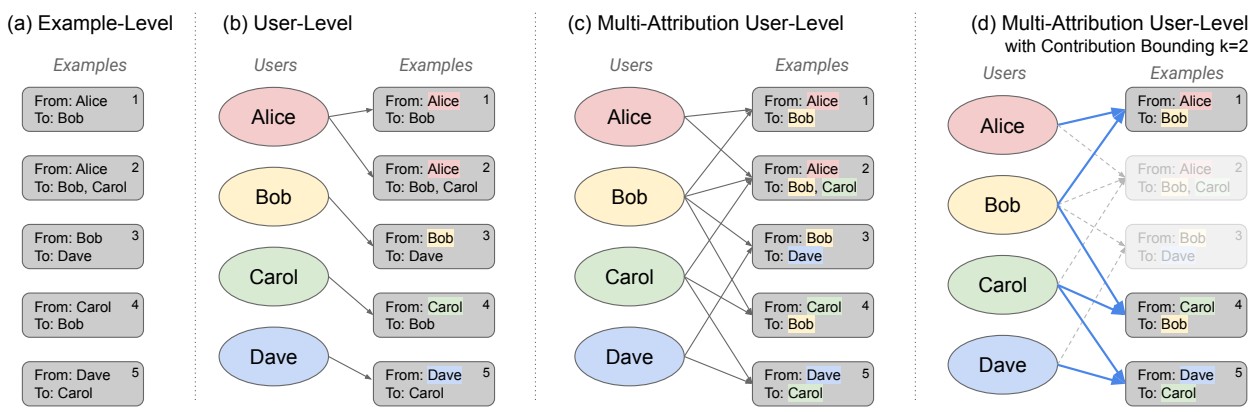

Figure 1. Different approaches to DP modeling for a toy email dataset. Each training example is the text of an email message. Viewing the dataset simply as a flat list of messages, (a), naturally gives rise to example-level DP. This may be too weak of a notion, as the same "secret" might occur in multiple emails. In (b), each email is attributed to the sender only, which leads to the standard (single-attribution) user-level DP notion. Again, this may not be sufficient if some secret from user Bob was contained both in emails he sent and received (for example, suppose he receives a message from Alice *"Hi Bob, I hope your recovery from surgery is going well."*). To address this issue, in Section 2 we use a hypergraph model of the dataset, (c): We attribute each example to the full set of senders and receivers, leading to the hypergraph with nodes (users) $V = (A, B, C, D)$ and edges (examples) $V = (e_1 = \{A, B\}, e_2 = \{A, B, C\}, e_3 = \{B, D\}, e_4 = \{C, B\}, e_5 = \{D, C\})$. With this structure, in (d) we can apply our contribution bounding algorithms (Section 4) to select a subset of the training examples $S$ so that each user contributes at most $k$ selected examples, for example selecting $S = \{e_1, e_4, e_5\}$ satisfies $k = 2$. After this pre-processing, user-level DP guarantees can be obtained using relatively standard ML infrastructure and DP algorithms.

*contained the same private information about you, that information is protected. Someone with access to the trained model would learn (nearly) the same thing even if the text of every email you ever sent or received was replaced with completely different text before training."*

**Contribution bounding algorithms:** Because of the overlap between data belonging to different users, we cannot use a user's average gradient as an example gradient in an algorithm like DP-SGD and immediately convert the resulting example-level DP guarantee into a user-level DP guarantee. Instead, we use *contribution bounding*: we find $S$, a subset of the data which each user contributes at most $k$ examples to, and then train on $S$. Given the contribution bound, it is well-known how to derive privacy guarantees for DP-SGD run on $S$ (see Section 3), so the overall procedure satisfies fixed-graph DP.

While in the single-attribution model contribution bounding is a trivial problem, in the multi-attribution model it is much more challenging. For example, maximizing $|S|$ while satisfying the contribution bound requirement is straightforward in the single-attribution model, whereas in the multi-attribution model this is an NP-hard problem (see Appendix B). In Section 4, we propose greedy baseline algorithms for contribution bounding which are the basis of the studies in the rest of the paper.

**Algorithmic choices:** In Section 5 we perform empirical evaluations to study a number of algorithmic design choices

in the contribution bounding problem:

*Unique vs. duplicate examples:* In Section 5.2 we study two variants of the greedy contribution bounding algorithm, which do or do not allow duplicate examples in $S$. Allowing duplicate examples gives a larger dataset with more uniform contributions among users, reducing the DP noise but potentially introducing bias.

*DP-SGD vs DP-MF:* DP-MF is a variant of DP-SGD which adds correlated noise across iterations. For example-level DP, it is known that DP-MF with privacy amplification captures DP-SGD with privacy amplification as a special case, and in turn Pareto dominates DP-SGD (Choquette-Choo et al., 2024). When moving to the multi-attribution model, it is not obvious how to do amplification for general DP-MF, whereas known amplification guarantees for DP-SGD still apply. Hence, it is not immediate which of the two algorithms should be preferred. We empirically compare the two in Section 5.3.

*Suboptimality of greedy algorithm:* Due to the hardness of maximizing $|S|$, a natural question is whether the greedy algorithm can be substantially improved on. For our research-scale datasets, in Section 5.4 we were able to solve an integer linear program (ILP) for this problem and compare the greedy algorithm to the optimal solution found by the ILP.

*Optimizing for bias:* Our contribution bounding algorithm biases towards examples attributed to fewer users. In Section 5.5 we study how impactful this bias is on the test

performance of our model, and variants of the greedy algorithm that attempt to mitigate this bias.

## 1.2. Related work

How to best fit an algorithm like DP-SGD to user-level DP was well-studied in the single-attribution setting by Charles et al. (2024); Chua et al. (2024). They show user-to-example reductions generally outperform algorithms that sample/batch at the example-level. However these reductions do not readily translate to the multi-attribution setting.

Outside learning contexts, the multi-attribution setting is analogous to node-level DP for hypergraph queries (we discuss this connection in more detail in Section 2). Node-level DP for graphs has been studied extensively (Blocki et al., 2013; Chen & Zhou, 2013; Kasiviswanathan et al., 2013; Borgs et al., 2015; Raskhodnikova & Smith, 2016a;b; Day et al., 2016; Borgs et al., 2018; Sealfon & Ullman, 2021; Kalemaj et al., 2023; Jain et al., 2024) and more recently has also been studied in hypergraphs (Fang et al., 2022).

## 2. Defining Privacy for Training Examples with Multiple Attribution

Given parameters $\varepsilon \geq 0$ and $\delta \in [0, 1]$, we say two distributions $P$ and $Q$ on the same space ($\sigma$-algebra) are $(\varepsilon, \delta)$-indistinguishable if, for all events $\mathcal{E}$,

$$P[\mathcal{E}] \leq e^{\varepsilon} Q[\mathcal{E}] + \delta \qquad \text{and} \qquad Q[\mathcal{E}] \leq e^{\varepsilon} P[\mathcal{E}] + \delta.$$

We denote this relation $P \approx_{\varepsilon, \delta} Q$. Abusing notation slightly, given two random variables $X, Y$ taking values in the same set, we write $X \approx_{\varepsilon, \delta} Y$ if their distributions satisfy the relation. We say a mechanism (that is, a randomized algorithm) $\mathcal{M}$ is $(\varepsilon, \delta)$-differentially private if for any pair of adjacent datasets $D$ and $D'$, we have $\mathcal{M}(D) \approx_{\varepsilon, \delta} \mathcal{M}(D')$. The definition of adjacency critically determines the "unit of privacy", and hence the strength and meaning of the protection offered at a particular $(\varepsilon, \delta)$. In this section we consider different definitions of DP that arise from different definitions of adjacency for datasets with multi-user training examples.

For simplicity, we specialize to machine learning applications with a training dataset consisting of examples $(x_1, \ldots, x_m)$. We associate with each training example $x_i$ additional metadata $e_i$ (not directly used in training). Formally, each $e_i$ is a subset of nodes (users) from some ground set $V$. We say the example $x_i$ is *attributed to* the nodes in $e_i$. Letting $E$ denote the list of $e_i$'s, we obtain a hypergraph $(V, E)$, possibly with repeated hyperedges (that is, multiple $x_i$ might be associated with the same subset $e \subseteq E$.). For instance, in an email dataset, $V$ could be a set of email addresses, account names, or other identifiers occurring in the data; each $x_i$ would be an email message, and $e_i$ would be a set of identifiers in $V$ to which the message is attributed—say, the sender and all the receivers of the message. For simplicity, we refer to the $u \in V$ as users.

In real datasets, the data curator must choose how to perform this attribution. We discuss this choice below; for now, we take it as a given. The input to a DP algorithm $\mathcal{M}$ is thus an attributed data set $D = ((x_1, e_1), \ldots, (x_m, e_m))$.

What notion of differential privacy makes sense for such data sets? We consider three notions here.

**Edge-data (Example-level) DP.** The simplest notion we consider is to change a single training example (that is, the data associated with a single hyperedge in the graph). We say two datasets $D, D'$ are *edge-data adjacent* if their associated hypergraphs are the same (that is, $m = m'$, and $e_i = e_i'$ for all $i \in [m]$), but differ by changing one example: $x_j \neq x_j'$ for a single index $j$. The algorithm $\mathcal{M}$ is *edge-data* $(\varepsilon, \delta)$-*differentially private* if, for all edge-data-adjacent data sets $D, D'$, we have $\mathcal{M}(D) \approx_{\varepsilon, \delta} \mathcal{M}(D')$.

Edge-data differential privacy is exactly the standard notion of differential privacy, applied to the list of examples in the dataset. In the email context, it protects the text of a single email but provides little guarantee about information that is common to many of a person's emails such as their religious beliefs or politics. This formulation of DP ignores the hypergraph altogether; it corresponds to Example (a) from Figure 1.

**Fixed-graph (multi-attribution user-level) DP.** In this paper, we consider a much stronger guarantee, which we call *fixed-graph* differential privacy. Our primary motivation is to derive practical DP training algorithms for hypergraph-annotated datasets that nevertheless protect a person's contributions to the data set as a whole.

**Definition 2.1.** We say two datasets $D, D'$ are *fixed-hypergraph adjacent* if their associated hypergraphs are the same and there is a user $u$ such that $x_i = x_i'$ for all indices $i$ such that $u \notin e_i$. That is, the examples involving one user (the $x_i$ where $u \in e_i$) may differ arbitrarily, but all examples not involving that user are the same.

A randomized algorithm $\mathcal{M}$ is *fixed-hypergraph* $(\varepsilon, \delta)$-*differentially private* if, for all *fixed-hypergraph adjacent* data sets $D, D'$, we have $\mathcal{M}(D) \approx_{\varepsilon, \delta} \mathcal{M}(D')$.

*Attribution and the choice of the hypergraph.* This definition of privacy separates the hypergraph (the attribution pattern) from the examples associated with hyperedges (the substance of the training data). The definition allows flexibility in how attribution is performed, which in turn determines the hypergraph attached to the training examples and, crucially, the interpretation of the DP bound. For example, in the case of emails, we could choose to attribute each email *only* to the sender (single attribution), so that each set $e_i \in E$ has

size 1. The resulting notion of DP corresponds directly to the typical application of user-level DP (e.g., McMahan et al. (2017)). Figure 1(b) illustrates this choice of hypergraph for an email setting.

However, for structured data like email, we can choose a hypergraph that leads to a much stronger[1] notion of privacy: by attributing it to the sender *and* all recipients as in Figure 1(c), we encode the possibility that the email contains sensitive information about any or all of these users[2].

**Node privacy.** Finally, one can consider a third guarantee that is stronger still. Since we consider datasets where users correspond to nodes, a natural notion of differential privacy arises by defining adjacency based on the addition or removal of a single node (see citations in Section 1.2).

**Definition 2.2** (Fang et al. (2022))**.** We say two datasets $D, D'$ are *node-adjacent* if there exists a vertex $u$ in $V'$ such that $D' = \{(x_i, e_i) \in D : u \notin e_i\}$ (or vice-versa, with $D$ and $D'$ swapped). A randomized algorithm $\mathcal{M}$ is *node* $(\varepsilon, \delta)$*-differentially private* if, for all *node-adjacent* data sets $D, D'$, we have $\mathcal{M}(D) \approx_{\varepsilon, \delta} \mathcal{M}(D')$.

This definition implies that the hypergraphs differ by a single node: that is, $V' = V \setminus \{u\}$ and $E' \subseteq E$ is the list of hyperedges (with repetitions) not including $u$.

Accurate node DP algorithms are hard to design. Even in the case of dyadic graphs (the "usual" graphs, in which each hyperedge connects exactly two nodes), most functions one would want to compute have extremely high worst-case sensitivity to the addition of a single node. As a result, considerable work has been devoted to designing node-DP algorithms that are accurate on "nice" graphs—for example, graphs with a given upper bound on nodes' degrees—yet provide the worst-case privacy guarantee of Definition 2.2.

Much less is known about designing node-private algorithms for hypergraphs. Even for the simple task of counting the number of hyperedges in the graph, existing algorithms (Fang et al., 2022) are based on solving a linear program with a variable for each edge in the graph. Training a language model on examples associated with hyperedges requires solving gradient averaging—a task mathematically similar to edge counting—once for each training step and parameter in the model. Current algorithmic approaches would not scale even to modest model sizes.

**On the choice to use fixed-graph DP:** *The algorithms in this paper satisfy fixed-graph DP.* The main weakness of fixed-graph DP in comparison to node DP is that since adjacent datasets share the same hypergraph, implicitly the hypergraph is public information. Of course, in many contexts the hypergraph itself is privacy-sensitive, *and we do not advocate for publishing the hypergraph in practice*. For example, the hypergraph might contain the fact that an individual received an email from a doctor who specializes in cancer treatment. This should be considered private information, regardless of the contents of this email. In light of this weakness, we offer two main justifications for our choice:

First, protecting the contents of interactions is still a substantial guarantee. For example, if the examples $x_i$ on hyperedges are the texts of emails, then the definition implies that the outputs would be (roughly) the same even if all the texts of Alice's emails were deleted from the data set. We additionally emphasize that fixed-graph DP in the multi-attribution setting is still a much stronger privacy protection than privacy protections in the now-standard single-attribution setting, which may altogether ignore the fact that multiple users' privacy can be impacted by a single example.

Second, while fixed-graph DP technically allows the hypergraph to be made public, we do not advocate for this and design algorithms that are unlikely to actually leak information about the hypergraph. In more detail, the specific algorithms we consider have a two-phase structure: they prune the hypergraph $H$ *without looking at the contents of the examples* $x_i$ to obtain a subgraph $H'$ that is "nice" (roughly, a hypergraph with a known maximum degree). They then run an algorithm that processes the training examples while ignoring the graph structure altogether. Since our uses of the graph structure and of the contents of examples are largely decoupled, any violations of the stronger notion of node privacy (Definition 2.2), i.e. any information leaked about $H$, can only come from correlations between the graph structure and training examples. Indeed, the only counterexamples to node-privacy we know are extremely brittle.

At a technical level, using fixed-graph DP allows us to sidestep the difficulties of designing truly node-private algorithms, which is that of bounding contributions in a stable or "insensitive" way. That is, unless substantial progress is made in the field of node DP for hypergraph data, we believe fixed-graph DP allows the minimal slack in the privacy definition needed to design private ML algorithms without completely sacrificing utility.

Our use of fixed-graph DP, rather than node DP, raises (at least) two open questions. First, is it possible to get node-DP algorithms with similar accuracy and efficiency to the algorithms we consider? It would require significantly new

---

[1]"Stronger" means the definition treats more datasets as adjacent; the protection at a fixed $(\varepsilon, \delta)$ is therefore higher.

[2]We assume throughout the paper a setting like training on emails where the metadata needed to define $E$ is readily available. In other settings $E$ may be challenging to materialize, e.g. when training on images containing faces that must be identified. In these settings it may be preferable to make additional assumptions and/or resort to the single-attribution model, but this is beyond the scope of this paper.

**Algorithm 1** The DP-SGD/DP-MF algorithms

**Inputs:** Dataset $D$, number of rounds $T$, batch size $B$, clip norm $C$, noise multiplier $\sigma$, initial model $\theta_0$, learning rate $\eta$, correlation matrix $\mathbf{C}$ (for DP-SGD, $\mathbf{C} = \mathbf{I}$).

1:  $\text{clip}(\mathbf{v}, C) := \mathbf{v} \cdot \min\{1, C/\|\mathbf{v}\|_2\}$
2:  **for** $i$ in $[T]$ **do**
3:      Sample a batch $\mathcal{B}_i \subseteq D$ of (expected) size $B$
4:      $\mathbf{g}_i \leftarrow \frac{1}{B}\sum_{\mathbf{x} \in \mathcal{B}_i} \text{clip}(\nabla\ell(\theta_{i-1}; \mathbf{x}), C)$
5:      $\mathbf{z}_i \sim N(0, \mathbf{I})$
6:      $\widetilde{\mathbf{g}}_i \leftarrow \mathbf{g}_i + \frac{C\sigma}{B}(\mathbf{C}^{-1}\mathbf{z})_i$          $\triangleright \mathbf{z} = (\mathbf{z}_1, \mathbf{z}_2, \ldots)$
7:      $\theta_i \leftarrow \theta_{i-1} - \eta\widetilde{\mathbf{g}}_i$ (or another first-order update)
8:  **end for**

algorithmic ideas. Second, do algorithms with the two-phase structure admit actionable attacks for realistic data sets? While we believe this is quite unlikely, it remains a possible and, we hope future work either strengthens or refutes this belief by trying to attack our fixed-graph DP algorithms. More broadly, we hope this work will spur further investigations of both models.

## 3. Multi-attribution DP-SGD/DP-MF

DP-SGD (Song et al., 2013; Bassily et al., 2014) is a canonical private learning algorithm which satisfies $(\varepsilon, \delta)$-DP. In DP-SGD, given in Algorithm 1, in each round $i$ we sample a batch $\mathcal{B}_i$ of examples from the dataset $D$ and compute their gradients on the current model, clip the gradients to have bounded $\ell_2$-norm, and add independent Gaussian noise to their average. We then use this as a noisy gradient in SGD, or another first-order method such as Adam.

DP-MF (matrix factorization) (Kairouz et al., 2021; Denisov et al., 2022; Choquette-Choo et al., 2022), also known as DP-FTRL and also given in Algorithm 1, is a variant of DP-SGD in which the noise used in different rounds is correlated, so that some of the noise added in one round is cancelled out in subsequent rounds. The amount of cancellation is specified by a lower triangular noise correlating matrix $\mathbf{C}^{-1} \in \mathbb{R}^{n \times n}$, which produces noise $(\mathbf{C}^{-1}\mathbf{z})_i = \sum_{j \le i} \mathbf{C}^{-1}_{i,j}\mathbf{z}_j$ in round $i$. The case $\mathbf{C}^{-1} = \mathbf{I}$ retrieves the independent noise of DP-SGD. The noise correlating matrix $\mathbf{C}^{-1}$ is the inverse of the *strategy matrix* $\mathbf{C}$; analysis of the strategy matrix is essential to choosing the correct noise multiplier $\sigma$ as discussed next in Section 3.1, but $\mathbf{C}$ does not appear directly in the implementation of the algorithm.

These two mechanisms typically require selecting minibatches in different ways and hence require different privacy analyses, which we discuss further below.

### 3.1. Minibatch selection and privacy accounting

We now discuss existing privacy accounting results we can repurpose for the multi-attribution model. In each case we state an already-proven privacy guarantee that protects any set of examples in $D$ that satisfy some property. To apply them to the multi-attribution setting, it then suffices to preprocess the graph dataset $(V, E)$ into a "nice" graph dataset $(V, E')$ such that sets of the form $\{e_i \in E : u \in e_i\}$, i.e. the sets of edges all including one user $u$, all satisfy the desired property[3], which will be the focus of Section 4.

**Poisson Sampling with DP-SGD:** DP-SGD typically chooses a batch $\mathcal{B}_t$ using Poisson sampling, where each example $x_i$ is independently included in $\mathcal{B}_t$ with probability $p = B/|D|$ where $B$ is a target batch size. Poisson sampling benefits from privacy amplification by sampling (Steinke, 2022), i.e. the randomness in the sampling amplifies the existing privacy guarantees. The example-level DP guarantee of this mechanism is well understood (Balle et al., 2018), and Charles et al. (2024) derive tight group-level $(\varepsilon, \delta)$-DP guarantees for this mechanism, i.e. guarantees assuming two adjacent datasets differ in at most $k$ examples. To apply them to multi-attribution datasets it then suffices to bound the max degree as $\max_{u \in V}|\{e_i \in E : u \in e_i\}| = k$. While DP-SGD's privacy guarantees have been studied in a number of other settings, this will be the main privacy accounting result we leverage for the rest of the paper. Since the privacy analysis is agnostic to how the noisy gradients are used, this analysis extends to e.g., DP-Adam and other variants.

**Min Separation with DP-MF:** There are several variants of DP-MF; in this work we restrict our focus to the banded matrix factorization (BandMF) mechanism (Choquette-Choo et al., 2024; McKenna, 2024). BandMF is characterized by a $b$-banded strategy matrix $\mathbf{C}$ ($\mathbf{C}_{i,j} = 0$ for $i - j \ge b - 1$) which determines the correlation structure; note that setting $b = 1$ gives independent noise / DP-SGD. BandMF's privacy analysis is easy when combined with a $(k, b)$-*minimum separation (min sep)* batch selection procedure. For our purposes, given batches $\mathcal{B}_1, \ldots, \mathcal{B}_T$, a set of examples satisfies $(k, b)$-min sep if examples in the set do not collectively appear more than once in any subsequence of consecutive $b$ batches $\mathcal{B}_t, \mathcal{B}_{t+1}, \ldots, \mathcal{B}_{t+b-1}$, and no more than $k$ times total across $\mathcal{B}_1, \ldots, \mathcal{B}_T$. The ordering here is generally deterministic and hence this batch selection strategy does not enjoy amplification guarantees[4]. Assume $\mathbf{C}$

---

[3]Note that other sets of examples may also satisfy the desired property, but this is immaterial to our desired fixed-graph DP guarantees.

[4]BandMF can additionally be combined with Poisson sampling to also benefit from privacy amplification, but doing so requires solving a hard clustering problem and potentially discarding a large fraction of the data in the multi-attribution setting, so we do

has fixed column norm (wlog) 1. Then if any pair of adjacent datasets differs only in a set of examples satisfying $(k, b)$-min sep, then DP-MF is a Gaussian mechanism (whose privacy guarantees are computable via results in (Balle & Wang, 2018)) with noise multiplier $\sigma/\sqrt{k}$ (Choquette-Choo et al., 2024). For fixed-graph DP, it then suffices if the sets of examples $\{e_i \in E : u \in e_i\}$ each satisfy $(k, b)$-min sep.

## 4. Contribution bounding algorithms

Given the results in Section 3, a natural algorithm for model training in the multi-attribution setting is to (i) compute a (multi-)subset of examples $S \subseteq D$, such that $S$ has at most $k$ examples per user (or if using DP-MF, such that after forming batches from $S$, each user's examples satisfy $(k, b)$-min sep), and then (ii) run DP-SGD/DP-MF on $S$. Under fixed-graph privacy, as long as we generate $S$ independently of the content of examples, we incur no privacy cost (in practice, of course, the hypergraph structure should still be kept confidential). Once we've chosen $S$, privacy accounting and training have minimal differences from the same problem in the single-attribution setting, which is well-understood. Hence, most of the challenge is choosing $S$. We refer to the problem of selecting $S$ as the *contribution bounding problem*, and this problem will be the main focus of the remainder of the paper.

A natural objective for the contribution bounding problem is to maximize $|S|$, since the DP noise multiplier reduces with a larger dataset size and the DP noise usually has a large impact on utility. However, in Appendix B we show that the problem of maximizing $|S|$ under a contribution bound constraint is NP-hard. Given this hardness, we instead propose a natural greedy algorithm with three variants. Due to space constraints, we give informal descriptions here and defer pseudocode to Appendix B.2. For each variant we first sort the dataset beforehand in increasing order of users per example.

**DP-SGD, no duplicates (Algorithm 2):** We take a single pass over the dataset, adding each example $(x_i, e_i)$ to $S$ if no user in $e_i$ has $k$ examples in $S$ already.

**DP-SGD, with duplicates (Algorithm 3):** Alternatively, we take multiple passes over the dataset, adding each example to $S$ if no user in $e_i$ has $k$ examples in $S$ already, and allowing duplicate examples to be added. We terminate after the first pass over $D$ that doesn't add any examples.

**DP-MF (Algorithm 4):** We take multiple passes over the dataset. $S$ is now a list, and we add each example (allowing duplicates) if after adding it, $S$ split into batches of size $B$ would not violate min sep of $b$ for any user's examples. We

not consider this approach in this paper. For completeness we give a discussion of privacy amplification for BandMF in Appendix A.

terminate when either we take a pass over $D$ without adding any examples (in which case the algorithm fails to find a solution), or when $S$ has length $TB$. We do not enforce a contribution bound requirement, but instead compute the contribution bound $k$ (which is at most $\lceil \frac{T}{b} \rceil$) post-hoc.

In Section 5 we will see that these baseline algorithms are already competitive, but there are many algorithmic choices one could make to further optimize the algorithm.

## 5. Empirical results

### 5.1. Experimental setup

We consider two tasks: training a small transformer on the arXiv dataset, and a synthetic logistic regression task.

**arXiv:** The arXiv dataset contains 1.9 million examples, each corresponding to a paper posted on arXiv. For each paper $i$, we take the set of users $e_i$ to be authors of the paper, and the content of each example $x_i$ to be the abstract of the paper. That is, we attribute each example to the authors of the corresponding paper. We use a train/test/validation split of 1.7M/0.1M/0.1M.

**Logistic regression:** To understand how generalizable our results on arXiv are, we also experiment on a synthetic logistic regression dataset.

*Generation of the synthetic graph:* Given the number of hyperedges $|E|$, the expected users per example $\mathbb{E}[|e|]$ and expected examples per user $d_u$, we sample $|E|$ random hyperedges, either under a probabilistic regular graph model or a correlated sampling model that induces a more skewed distribution of examples per user. We defer details of the sampling procedures to Appendix C.1. We tried graph sizes $|E| \in \{125k, 1M, 8M\}$, expected examples per user $d_u \in \{2, 4, 8\}$, fix $\mathbb{E}[|e|] = 2$, and consider both regular and skewed graphs.

*Generation of the synthetic logistic regression data:* Each hyperedge $e_i$ has data $x_i = (z_i, y_i)$ with features $z_i \in \mathbb{R}^d$ (sampled from a multivariate normal) and label $y_i \in \{0, 1\}$. The groundtruth in the data generation process consists of two components: a universal base vector $a$ which is shared across all samples, and a bias vector $b$ scaled by the number of users $|e_i|$ for each hyperedge. This recreates a form of sampling bias present in the arXiv dataset; see Section 5.5 for more discussion. We sample $y_i \sim$ Bernoulli(sigmoid($\langle a + |e_i|b, z_i \rangle$)). Further details about this procedure are deferred to Appendix C.1. We use a train/test/validation split of 80%/10%/10%.

**Models, and training with DP:** For arXiv, we fine-tune a pre-trained BERT-Tiny model for the masked language modeling task (Devlin et al., 2019), using the default BERT

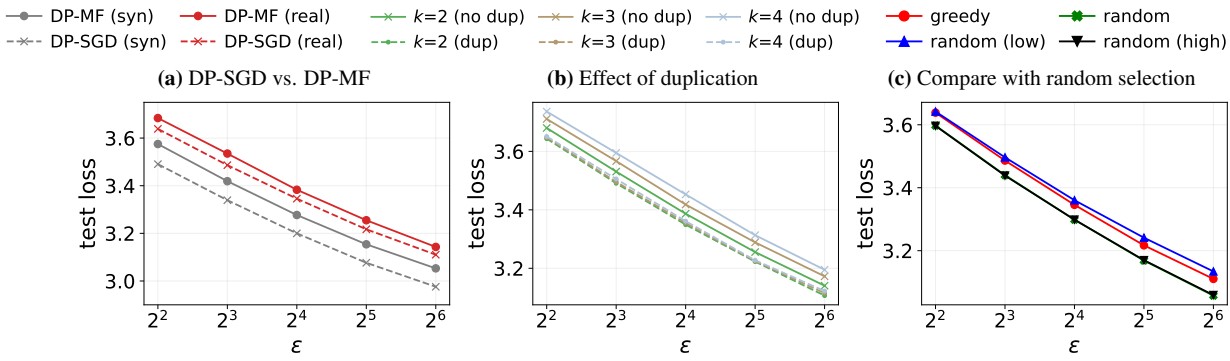

*Figure 2.* **Results on the arXiv dataset.** We plot the test loss as a function of $\varepsilon$.

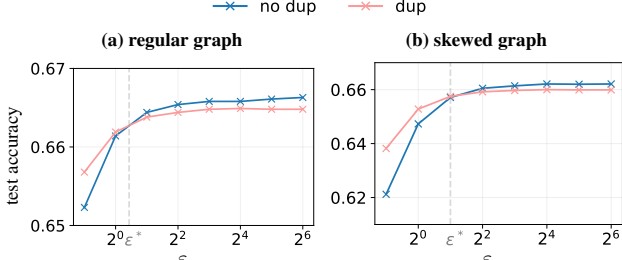

*Figure 3.* **Impact of duplication in the regression task.**

tokenizer and sequence length of 512. The training objective is cross entropy loss for masked token prediction. We report the prediction *loss* of the final model on the test set at $\varepsilon \in \{2^2, 2^3, \ldots, 2^6\}, \delta = 10^{-10}$. The pretrained checkpoint we use as an initialization has loss $4.955$, far larger than that of any of the private models we train. Due to space constraints, further details of training are deferred to Appendix C.2. For the logistic regression task, we fit a linear model with weight $w \in \mathbb{R}^d$ and bias $b \in \mathbb{R}$. The training objective is cross-entropy loss. We report the prediction *accuracy* of the final model (using threshold 0.5) on the test set at $\varepsilon \in \{2^{-3}, 2^{-2}, \ldots, 2^6\}, \delta = 10^{-10}$.

### 5.2. Duplicates vs no duplicates

For DP-SGD we have two variants of the contribution bounding algorithm, one that allows duplicates and one that does not. Allowing duplicates allows us to increase $|S|$ while satisfying the same contribution bound, in turn reducing the DP noise, but this comes at the cost of potentially increasing the sampling bias of $S$.

To understand the benefits and detriments of duplication, as a simple example, suppose our users are $\{A, B, C, D\}$ and our examples are associated with hyperedges $\{(A), (A, B), (A, C), (B, C), (A, D)\}$. If our contribution bound is $k = 3$, without duplicates we can get a dataset of size at most 4, since we have to discard one of the hyperedges including $A$. However, with duplicates we can use the dataset $\{(A), (A, B), (A, D), (B, C), (B, C)\}$

which includes 5 examples and still satisfies the bound $k = 3$. Since we have more examples at the same contribution bound training on the duplicated dataset will require less noise for DP, but it may also be overly biased towards the hyperedge $(B, C)$.

In Figure 2(b) we plot the test loss for arXiv, comparing the two algorithms across a sweep of different contribution bounds. For the arXiv experiment, duplication is consistently better than no duplication across all $\varepsilon$ and contribution bound values, suggesting for this task noise reduction is much more important than bias reduction. In Figure 3 we do the same for the logistic regression experiment. Here we see a threshold for the $\varepsilon$ value, $\varepsilon^\star$, where duplication helps only at $\varepsilon < \varepsilon^\star$. For smaller $\varepsilon$, noise is larger and thus it is more important to reduce than bias. We observe a larger $\varepsilon^\star$ in the skewed graph compared to a regular graph. We believe this is because for a skewed graph, under a fixed contribution bound $k$ more users will have $< k$ examples than for the regular graph, so the size of the dataset under contribution bound $k$ will be smaller and hence the DP noise needed will be larger. As a simple example, in a $k$-regular graph we can include all examples, but in a graph where half the users' degrees are 1 and the other half's are $2k - 1$, we will need to remove a constant fraction of the examples to satisfy the contribution bound. We will revisit the bias-variance tradeoff in Section 5.5.

### 5.3. DP-SGD vs DP-MF

We next compare the DP-SGD and DP-MF noise addition algorithms and their corresponding greedy contribution bounding algorithms (based on the previous section, for DP-SGD we focus on the contribution bounding algorithm that allows duplicates). Recall that in e.g. the example-level DP setting, DP-SGD with privacy amplification and its privacy analysis is a special case of DP-MF with privacy amplification and its analysis (Choquette-Choo et al., 2024). In the multi-attribution case, the set of known amplification results are limited (see Appendix A) and we believe it is an important open question to give tight amplification guarantees for

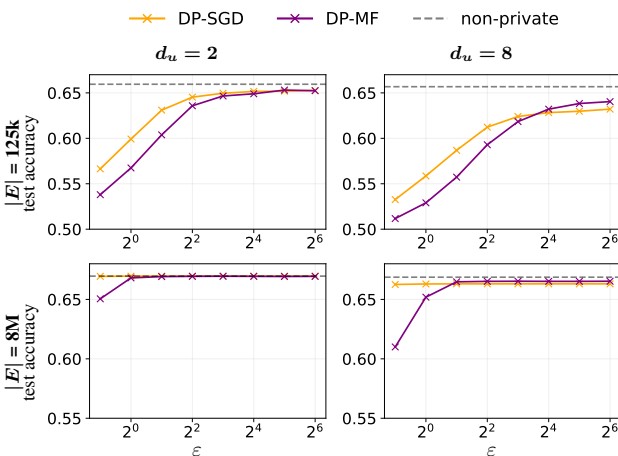

Figure 4. **Comparing DP-SGD vs. DP-MF in regression.**

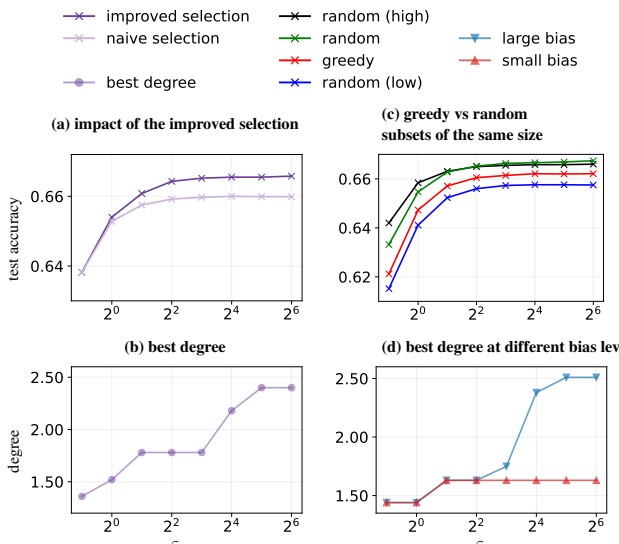

Figure 5. **Experiments on data bias in the regression task.**

user-level DP-MF with example-level sampling (in both the single- and multi-attribution case). We emphasize that if we had tight guarantees, DP-SGD in the multi-attribution setting and its analysis would also be a special case of DP-MF and its analysis, and one could default to DP-MF in all settings. However, in absence of such tight guarantees, we are required to choose between DP-SGD with privacy amplification and DP-MF without privacy amplification.

In Figure 2(a) we compare the relative performance of DP-SGD and DP-MF at different $\varepsilon$ on the arXiv dataset. We observe that DP-SGD outperforms DP-MF at all $\varepsilon$. We additionally observe that this holds even after we replace the real arXiv graph by a randomly generated regular hypergraph. We believe this is in part because our tuned parameters use a small number of iterations per epoch, which reduces the min sep $b$. DP-MF benefits from larger min sep (Choquette-Choo et al., 2024) and hence the advantages of DP-MF are minimized in this setting.

In Figure 4 we compare DP-SGD and DP-MF's test accuracy on logistic regression for different instantiations of the synthetic dataset. Across the various dimensions we examined (graph structure and size, examples per user, and $\varepsilon$ values), we identify scenarios where DP-MF can have an edge over DP-SGD. The advantage becomes more evident when: (i) $\varepsilon$ is larger, (ii) the graph structure is more skewed (heterogeneous users), or (iii) when users on average attribute more samples. The improvement of DP-MF in (i) is due to amplification being weaker at higher $\varepsilon$ and is consistent with comparisons in past work (Choquette-Choo et al., 2024). We believe the improvement of DP-MF in (ii) is because skewed graphs lead to a higher percentage of duplicates in $S$ for our DP-SGD contribution bounding algorithm (which will only add these duplicates for users with few examples), which increases bias. The DP-MF contribution bounding algorithm does not enforce a hard contribution bound, i.e. whatever duplicates it adds may be distributed

more uniformly across the examples, which would mitigate this bias. The improvement of DP-MF in (iii) can be explained by the fact that high-privacy-loss events where a user contributes multiple times to the same batch are more frequent in this setting, but DP-MF's min sep condition avoids this possibility altogether.

### 5.4. Greedy vs ILP

We next study the suboptimality of the contribution bounding algorithm. We write and solve an integer linear program to find exact solutions for the contribution bounding problem. For arXiv without duplicates, the greedy algorithm only selects $\approx 1.3\%$ fewer samples than the ILP, which translates to a negligible difference in test performance. This is evidence that the hardness of the contribution bounding problem is mostly pathological, i.e. there is limited room to improve training by trying to tackle the underlying combinatorial optimization problem. With duplicates the ILP actually includes fewer distinct examples. We give details on the ILP and results in Appendix C.4.

For DP-MF, we can again write an ILP, given in Appendix C.4. However, this ILP requires many more variables than our ILP for DP-SGD, so our solver did not finish in any of our empirical settings. It remains an interesting problem to determine the suboptimality in practice of the greedy algorithm for the DP-MF contribution bounding problem.

### 5.5. Bias vs num examples

Any contribution bounding algorithm that aims to maximize $|S|$ is likely to bias towards examples with fewer users. We first study the impact of this bias on training performance, and then study algorithms which attempt to mitigate this bias at the cost of including fewer examples.

Our initial experiment involves selecting random sets of examples of the same size as the solution arrived at by the greedy algorithm, disregarding the consideration for contribution bounding at all. That is, if $S$ is the solution arrived at by the greedy algorithm, we sample a random subset of the dataset $D$ of size $|S|$, $S'$. We then compare training on $S$ and $S'$ using the same noise multiplier, even though $S'$ might violate the contribution bound and thus not satisfy the same privacy guarantees. In this way, the only difference between the test performance after training on $S$ instead of $S'$ is due to the bias in example selection. In addition to choosing $S'$ by sampling a random subset of $D$ as a whole, we consider sampling a random subset of examples which have $\leq u$ (called "random (low)") or $> u$ users (called "random (high)") where $u$ is the median of $\{|e_i| : e_i \in E\}$.

In Figure 2(c) and Figure 5(c) we compare the test performance of the greedy algorithm and different random selection strategies. For arXiv, we see that the greedy algorithm performs noticeably worse than training on a random subset, with the increase in test loss comparable to decreasing the privacy parameter $\varepsilon$ by a multiplicative factor of $\sqrt{2}$ or more. A random subset of examples with $> u$ users achieves as good or better performance than a random subset of all data, i.e., examples with more users tend to be higher quality. We observe similar comparisons for logistic regression, suggesting our bias term $b$ in the synthetic data generation captures the phenomenon of examples with more users being higher quality reasonably well.

Because the impact of bias is significant, we consider a variant of our greedy contribution bounding algorithm which mitigates the bias. Our algorithm splits the dataset into two groups, examples $D_s$ which have $\leq u$ users and examples $D_\ell$ which have $> u$ users, for some $u$. Rather than take a pass over the whole dataset, we take a pass over $c_1$ examples in $D_s$ followed by $c_2$ in $D_\ell$, alternating between the two. The choice of $u, c_1, c_2$ can be tuned to tradeoff average examples per user (i.e., bias mitigation) and dataset size. We describe the tuning procedure in Appendix C.5. We then examine the learning outcome of these various datasets of different degrees and sizes.

Experiments on logistic regression shows that carefully picking a sweetspot in the degree-size tradeoff can lead to non-negligible improvement compared to the naive greedy solution. In Figure 5(a-b) we plot the improvement in test accuracy due to optimizing this tradeoff, and the average degree of the dataset optimizing this tradeoff. As an additional comparison, in Figure 5(d) we reproduce Figure 5(b) but after generating datasets with $b$'s magnitude increased/decreased by $10\times$. In contrast, for arXiv we found that for the full range of $\varepsilon$ spanned by our experiments, the tradeoff was optimized by maximizing the dataset size, i.e. not mitigating

the bias at all. In conjunction with the results in Section 5.2, this suggests that for transformer training reducing the DP noise should be the primary goal in contribution bounding.

## Acknowledgements

We are thankful to: Zachary Charles, for sharing preprocessing utilities for the arXiv dataset. Peter Kairouz for feedback on an initial draft of this work. Vincent Cohen-Addad, Alessandro Epasto, Morteza Zadimoghaddam, Galen Andrew, and Amlan Chakraborty for discussions on scalable algorithms for contribution bounding. Thomas Steinke, for discussions on definitions of privacy for network data. The anonymous reviews for useful feedback on the initial submission.

## Impact Statement

Since we are advocating for model training under stricter privacy definitions, we anticipate the societal impact of this paper (as with most other papers in privacy-preserving machine learning) is strictly beneficial and does not need to be discussed in detail here.

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

# A. Poisson Sampling with DP-MF

**Cyclic Poisson Sampling with DP-MF**    Recall BandMF is characterized by a $b$-banded strategy matrix $\mathbf{C}$ ($\mathbf{C}_{i,j} = 0$ for $i - j \geq b - 1$) which determines the correlation structure; setting $b = 1$ gives independent noise / DP-SGD. BandMF can be combined with Poisson sampling, and since BandMF generalizes DP-SGD, in the single-attribution setting this gives a strict generalization of DP-SGD with Poisson sampling, i.e. this mechanism is never worse and often better (Choquette-Choo et al., 2024). However, the privacy guarantee of this mechanism requires a very specific minibatch sampling strategy. Specifically, we must partition our users into $b$ groups, for each group collect the subset of examples $D_0, \ldots, D_{b-1}$ *only* attributed to users in that group, then cyclically iterate through subsets and Poisson sample from each one (i.e. in round $i$, we subsample from $D_{i \pmod b}$). This sampling strategy endows the mechanism with amplification guarantees, enabling it to Pareto-dominate DP-SGD and other variants of DP-MF. This sampling strategy is possible in our multi-attribution setting, and the noise multiplier required to satisfy $(\varepsilon, \delta)$-DP can readily be computed with the same function used for DP-SGD (DP Team, 2022; Choquette-Choo et al., 2024). The main drawback of this approach (unique to the multi-attribution setting) is that examples attributed to users belonging to different groups are discarded, which may comprise a large fraction of the dataset. Deciding how to partition the users to minimize the amount of discarded data is a generalization of the NP-hard $k$-cut problem, adding even more complexity to our contribution bounding strategy. So, we do not consider this approach in our work and leave an exploration of the effectiveness of this approach in the multi-attribution setting to future work.

# B. Deferred Details from Section 4

## B.1. NP-Hardness of Contribution Bounding

Given an independent set instance, we can create an example for each vertex, and a user for each edge. Each example is attributed to each user corresponding to the edges adjacent to the example's vertex. Setting $k = 1$, any solution to the contribution bounding problem corresponds to an independent set of the same size. Hence, the contribution bounding problem (when the objective is to maximize $|S|$) is NP-hard even for $k = 1$, and e.g. even to approximate to within a constant factor, or if each example is attributed by at most 3 users (corresponding to independent set on bounded-degree graphs) (Berman & Furer, 1994).

## B.2. Pseudocode for Algorithms

---

**Algorithm 2** Greedy contribution bounding algorithm for DP-SGD (without duplicates)

**Inputs:** Dataset $D$ with attribution graph $G$, contribution bound $k$

1: Sort examples in $D$ by increasing cardinality of corresponding edge in $G$.
2: $S \leftarrow \emptyset$
3: **for** example $x$ in $D$ **do**
4:     **if** $S \cup \{x\}$ satisfies contribution bound $k$ for all users **then**
5:         $S \leftarrow S \cup \{x\}$
6:     **end if**
7: **end for**
8: **return** $S$

---

---

**Algorithm 3** Greedy contribution bounding algorithm for DP-SGD (with duplicates)

---

**Inputs:** Dataset $D$ with attribution graph $G$, contribution bound $k$

1: Sort examples in $D$ by increasing cardinality of corresponding edge in $G$.
2: $S \leftarrow \emptyset$                                                        ▷ empty multi-set
3: **for** example $x$ in $D$ (repeated) **do**
4:      **if** $S \cup \{x\}$ satisfies contribution bound $k$ for all users **then**
5:          $S \leftarrow S \cup \{x\}$
6:      **end if**
7:      **if** No examples were added to $S$ since the last time we processed $x$ **then**
8:          **return** $S$
9:      **end if**
10: **end for**

---

**Algorithm 4** Greedy contribution bounding algorithm for DP-MF

---

**Inputs:** Dataset $D$ with attribution graph $G$, iterations $T$, batch size $B$, min sep parameter $b$

1: Sort examples in $D$ by increasing cardinality of corresponding edge in $G$.
2: $S \leftarrow []$                                                     ▷ empty list
3: **while** $|S| \leq TB$ **do**
4:      $x \leftarrow$ next example in $D$ (repeated)
5:      **if** $S \cup \{x\}$ satisfies $b$-min sep with batch size $B$ **then**
6:          $S \leftarrow S \cup \{x\}$
7:      **end if**
8:      **if** No examples were added to $S$ since the last time we processed $x$ **then**
9:          **return** No solution found
10:      **end if**
11: **end while**
12: **return** $S$

---

# C. Deferred Details and Results from Section 5

## C.1. More Details on Synthetic Data Generation

For our synthetic hypergraphs, we use two sampling procedures. Recall we start with a target number of hyperedges $|E|$, expected users per example $\mathbb{E}[|e|]$ and expected examples per user $d_u$.

**Probabilistic regular graph** For this graph, we assign each edge to each of the $m$ users with probability $\mathbb{E}[|e|]/m$. This leads to a binomial sampling process where the expected edge arity is $\mathbb{E}[|e|]$ as desired. To improve over the computational efficiency of this binomial sampling, we note that $m$ is large and $\mathbb{E}[|e|]/m$ is very small, enabling us to approximate this binomial distribution with a Poisson distribution. Concretely, for each edge $e$, we: 1) *Sample the edge arity $n_e$* from Poisson($\mathbb{E}[|e|]$); 2) *Choose $n_e$ users* uniformly at random from the $m$ total users. This two-step approach preserves the desired average edge arity of $\mathbb{E}[|e|]$ while being more efficient than the exact binomial sampling.

**Skewed graph:** In this graph, we attribute edges to users in a sequential manner. After determining the number of users and expected edge arity $\mathbb{E}[|e|]$, let $d_u(j)$ denote the current degree of user $j$. For the $i$th edge, we attribute it to user $j$ with probability $\frac{\mathbb{E}[|e|](1+d_u(j))^\alpha}{\sum_{j'}(1+d_u(j'))^\alpha}$. That is, users who already have edges attributed to them are slightly more likely to have future edges attributed to them as well, but we maintain that the average edge arity is $\mathbb{E}[|e|]$ throughout. The parameter $\alpha$ controls the degree of skewness, with larger $\alpha$ leading to a more skewed graph. We set $\alpha = 1.5$ in our experiment. We present a histogram of the number of edges owned by users in Figure 6, confirming that the graph is indeed highly skewed.

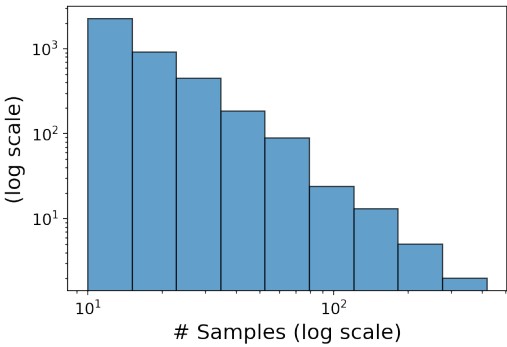

*Figure 6.* Histogram of the number of edges (or samples) owned by a user for a skewed graph with $|E| = 125\text{k}, d_u = \mathbb{E}[|e|] = 2$.

**Logistic regression data:** Each hyperedge $e$ is associated with a feature $z_i \in \mathbb{R}^d$ (sampled from a multivariate normal) and a label $y_i \in \{0, 1\}$. The groundtruth in the data generation process consists of two components: a universal base vector $a$ which is shared across all samples, and a bias vector $b$. Both are sampled from a multivariate Gaussian distribution with covariance $\frac{1}{\sqrt{d}}\mathbb{I}$ such that the average squared $\ell_2$-norm of a vector from this distribution is 1. For each example, features $z_i$ are sampled randomly from the same multivariate Gaussian, and the label $y_i$ is thus obtained by randomly sampling from a Bernoulli distribution, $y_i \sim \text{Bernoulli}(\text{sigmoid}(\langle a + b \cdot |e_i|, z_i \rangle; k_1))$, where $\text{sigmoid}(x; k_1) = \frac{1}{1+\exp(-k_1 x)}$ denotes the sigmoid function with steepness parameter $k_1$. We adopt $k_1 = 20$ in all experiments.

To further control how strongly the bias $b$ influences the labels (for the study in Section 5.5), we introduce a scaling transformation that replaces $a + |e_i|b$ with $\frac{2}{1+k}(a + k|e_i|b)$. The parameter $k$ (defaulting to 1) amplifies or diminishes the effect of the bias term, so larger values of $k$ place more emphasis on $b$, while smaller values mitigate its impact.

## C.2. More Details on Hyperparameter Tuning

We use Adam as our optimizer. For all experiments we sweep over the following parameter choices:

- Learning rate: $1 \times 10^{-4}, 2 \times 10^{-4}, 5 \times 10^{-4}, 1 \times 10^{-3}, 2 \times 10^{-3}, 5 \times 10^{-3}$.

- Clipping norm: $1, 0.5, 0.2, 0.1, 0.05, 0.02, 0.01$.

- Batch size and number of iterations: We keep the product of these fixed. For experiments on arXiv, the fixed product is chosen as $20, 480, 000 = 2^{11} \cdot 10^4$ and we vary the batch size in $2^{11}, 2^{12}, 2^{13}, 2^{14}, 2^{15}, 2^{16}$; for experiments on linear regression, the fixed product is chosen as $10, 240, 000 = 2^{10} \cdot 10^4$ and we vary the batch size in $2^{10}, 2^{11}, 2^{12}, 2^{13}, 2^{14}, 2^{15}$.

For DP-MF, we sweep the min sep parameter from $b = 2$ to the smallest value for which Algorithm 4 fails to find a solution, minus one.

We use the accounting procedures described in Section 3. We use a zero-out (Ponomareva et al., 2023) variant of the adjacency definitions in Section 2 for accounting, although this only changes the $\sigma \to \varepsilon$ mapping and all our observations still hold for arbitrary replacement as well. For arXiv, to understand the behavior of these algorithms in a practical setting with much more available compute (which usually leads to reduced noise; see Figure 1 of (Ponomareva et al., 2023) for more discussion), we employ a standard research practice of computing the noise multiplier assuming a hypothetical batch size that is larger than the physical batch size used for training. Concretely, we adopt a hypothetical batch size of $2^{18}$ for physical batch ranging from $2^{13}$ to $2^{16}$ and a hypothetical batch size of $2^{17}$ for physical batch ranging from $2^{10}$ to $2^{12}$. The practice of assuming larger batch/dataset sizes for accounting has been previously used by Charles et al. (2024).

## C.3. More Details on Contribution Bounding for Different Graphs

We apply the greedy contribution bounding algorithm (Algorithm 3) to the synthetic graphs, varying in the graph structure, graph size, and expected examples per user. We present the size of the resulting contribution bounded datasets in Figure 7.

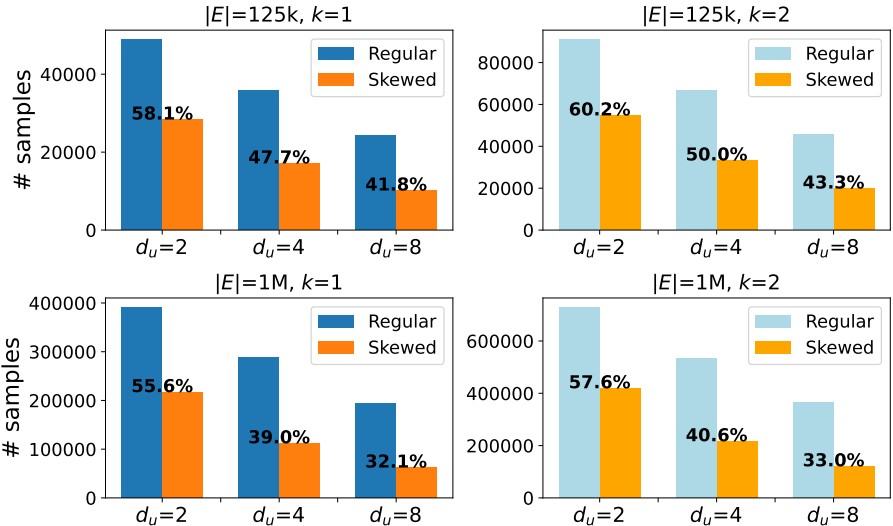

*Figure 7.* **Size of resulting datasets obtained via contribution bounding (Algorithm 3)**, contrasting a *regular* graph with a *skewed* graph, across different graph size $|E|$ (rows), contribution bounding parameter $G_{\text{ELS}}$ (columns), and expected examples per user $d_u$ (groups within a subfigure). The annotated percentages indicate the proportion of retained dataset size for the skewed graph relative to the regular graph.

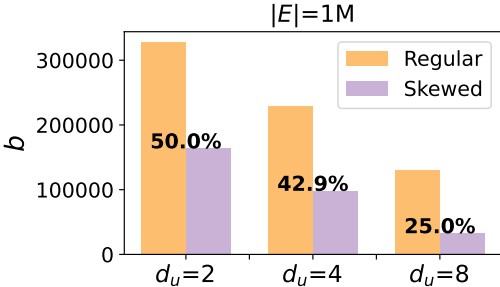

*Figure 8.* **The maximum feasible $b$ when applying greedy contribution bounding for DP-MF (Algorithm 4)** on *regular* vs. skewed graphs, across different expected examples per user $d_u$.

The key takeaways are as follows. First of all, skewness presents additional challenges to the selection problem. Second, the negative impact of skewness gets amplified when: 1) data size increase (comparing vertically) or 2) user owns more samples (comparing horizontally in each subfigure).

Experiments on greedy contribution bounding for DP-MF (Algorithm 3), as presented in Figure 8, points to similar conclusions.

## C.4. More Details on Greedy vs ILP Selection

While the contribution bounding problem is NP-hard even to approximate, for our research-scale datasets, we are able to formulate the contribution bounding problem as an integer linear program (ILP) and solve this ILP (for small values of $k$) to determine the suboptimality of the greedy algorithm. For DP-SGD, the contribution bounding problem is equivalent to the following ILP:

$$\max \sum_{e_i \in E} x_i : \qquad \forall j : \sum_{i:j \in e_i} x_i \leq k, \qquad \forall i : x_i \in \{0, 1\}. \tag{1}$$

Here, each index $i$ represents an example and each index $j$ represents a user, and $x_i$ is a variable which is 1 if we include

example $i$ in the subset $S$. The objective is to maximize the number of examples included, and the constraint $\sum_{i:j\in e_i} x_i \leq k$ enforces that user $j$ doesn't contribute more than $k$ examples. If we are allowing duplicate examples, we can replace $x_i \in \{0, 1\}$ with the constraint $x_i \in \mathbb{Z}_{\geq 0}$.

We solve the ILP using the open-source solver GLOP (Google, 2024) and compare the number of examples found by solving the ILP and by our greedy algorithm (without duplicates). In Figure 9 we first compare the number of examples chosen by both methods when run on the arXiv dataset, without allowing duplication. Because our solver did not finish for $k > 3$, we only give results for the settings $k = 2, 3$.

| Method | Contribution bound | Examples selected |
|--------|--------------------|-------------------|
| Greedy | 2 | 370019 |
| ILP | 2 | 374663 |
| Greedy | 3 | 475831 |
| ILP | 3 | 482128 |

*Figure 9.* Examples selected by the greedy algorithm and the ILP from the arXiv dataset.

We see that for both settings of $k$ the greedy algorithm retrieves $\approx 98.7\%$ of the ILP solution value, suggesting the hardness of the combinatorial optimization problem has a minimal impact in practice. In Figure 10 we next show that this translates to minimal degradation in the noise multiplier and test loss. For the noise multiplier, greedy achieves at worst a $1.25\%$ larger noise multiplier than the ILP solution at all $\varepsilon$ values, and for the test loss the increase is at most $.25\%$.

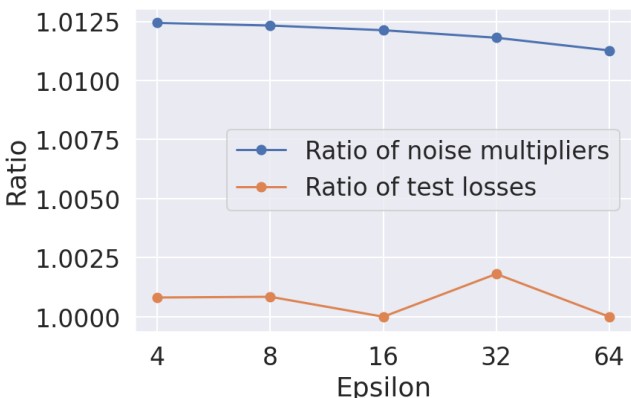

*Figure 10.* Ratio of the noise multiplier/test loss achieved by greedy versus ILP

When allowing duplicate examples, the ILP finds more examples, but fewer distinct examples as shown in Figure 11, leading to worse training outcomes. This is because the ILP can choose solutions which duplicate single-author examples many times, whereas the greedy algorithm will have to take a pass over the entire dataset and try adding all examples before it is allowed to pick its first duplicate, which limits its ability to overselect examples with few authors.

| Method | Contribution bound | Examples selected | Distinct examples selected |
|--------|--------------------|-------------------|----------------------------|
| Greedy | 2 | 442902 | 370019 |
| ILP | 2 | 462001 | 231335 |
| Greedy | 3 | 651409 | 475831 |
| ILP | 3 | 692943 | 231429 |

*Figure 11.* Examples selected by the greedy algorithm and the ILP from the arXiv dataset when duplication is allowed.

We also state an ILP for forming batches with $(k, b)$-min sep for completeness. The ILP is as follows:

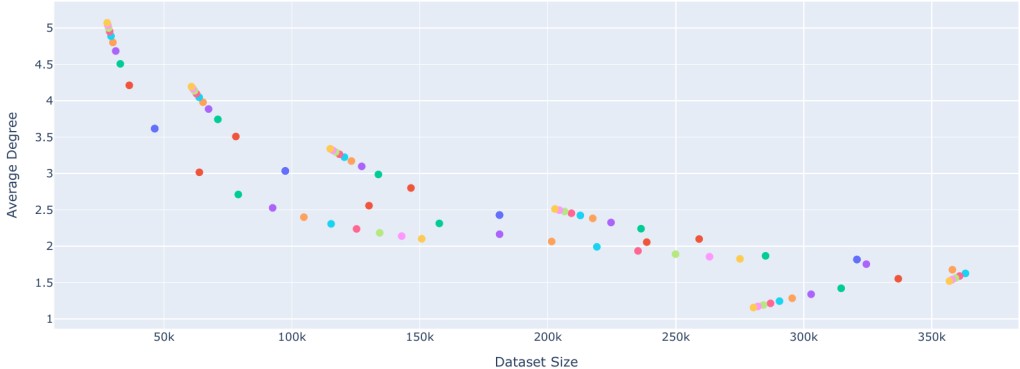

*Figure 12.* The dataset size and average edge degree of the 80 datasets generated by our bias-mitigating variant of Algorithm 2.

$$\min \quad k$$

$$s.t. \forall j, t \in [T - b + 1] : \sum_{i:j \in e_i} \sum_{t'=t}^{t+b-1} x_{i,t'} \leq 1,$$

$$\forall j : \sum_{i:j \in e_i} \sum_{t \in [T]} x_{i,t} \leq k,$$

$$\forall t : \sum_{e_i \in E} x_{i,t} = B,$$

$$\forall i : x_i \in \{0, 1\}.$$

The objective is to minimize the contribution bound $k$. Each example $i$ and time step $t$ has variable $x_{i,t}$ which is 1 if example $i$ is included in the $t$-th batch. The constraint $\sum_{i:j \in e_i} \sum_{t'=t}^{t+b-1} x_{i,t'} \leq 1$ says that user $j$ does not have more than one example attributed to them in batches $t$ to $t + b - 1$, i.e. the $b$-min sep condition holds for these batches. The constraint $\sum_{i:j \in e_i} \sum_{t \in [T]} x_{i,t} \leq k$ ensures each user's contribution bound is at most $k$. The constraint $\sum_{e_i \in E} x_{i,t} = B$ enforces that the $t$-th batch has exactly $B$ examples.

### C.5. Details on Bias vs Num Examples Experiments

As described in Section 5.5, we consider a variant of Algorithm 2 that splits the dataset into two groups, "small" examples $D_s$ which have $\leq u$ users and "large" examples $D_\ell$ which have $> u$ users, for some $u$. The algorithm takes a pass over $c_1$ examples in $D_s$ followed by $c_2$ in $D_\ell$, alternating between the two.

For these experiments we fix a contribution bound of 3. We sweep the threshold $u \in \{1, 2, 3, 4\}$, and consider $c_1 = 1, c_2 \in \{1, 2, \ldots 10\}$ and $c_1 \in \{1, 2, \ldots 10\}, c_2 = 1$. This gives a grid of 80 hyperparameter settings which each generate a dataset. We plot the 80 datasets' sizes and average edge degree (users per example) in Figure 12. We observe that some datasets have both smaller dataset size and average edge degree than another dataset. Hence for training, we only consider datasets that are not Pareto dominated in these two dimensions. In other words, for the datasets we do training on, the average degree is monotonically decreasing in their dataset size, and we can report either quantity alone as a proxy for a point on the tradeoff curve between these two quantities.

