# OpenReview forum: "It's My Data Too: Private ML for Datasets with Multi-User Training Examples"
_ICML.cc/2025/Conference — ICML 2025 poster_

### Official Review · Reviewer_FPm2 · 2025-03-13

**Overall Recommendation:** 3

**Summary:**

This paper discusses the user-level privacy under the multi-attribution scenario, where each user is associated with multiple examples but also each example can be attributed to multiple users. It starts with defining the differential privacy used in this case as the fixed-graph DP. Then it goes to discuss how to use some of the known algorithms, DP-SGD and DP-MF for the purpose of multi-attribution data. The paper discusses methods to choose the set S which is a subset of the data where each user contributes at most k examples. This subset needs to be maximized while keeping this restriction.
## update after rebuttal
I keep my score.

**Claims And Evidence:**

The paper in general follows well and is understandable, however, some claims raise some questions. I list them below:
- The claim that allowing duplicates allows for a larger dataset which reduces DP noise. I fail to understand why having duplicates leads to a larger dataset, as we would still have the same limit on contribution.
- In section 5.2, the authors hypothesize that the skew in the graph is because k more users will have <k examples than for the regular graph and then the dataset will be smaller. Again, I fail to see why this is true.

**Essential References Not Discussed:**

One reference I found missing is Abadi, M., Chu, A., Goodfellow, I., McMahan, H. B., Mironov, I., Talwar, K., & Zhang, L. (2016, October). Deep learning with differential privacy. In Proceedings of the 2016 ACM SIGSAC conference on computer and communications security (pp. 308-318).‏
When discussing the DPSGD algorithm, it makes sense to cite this work.

**Experimental Designs Or Analyses:**

The experiments are sound and valid as far as I can tell, although I do have the questions I mentioned in the claims and evidence section.

**Methods And Evaluation Criteria:**

The proposed methods and metrics give a clear idea of the advantages of the used method. Also, the comparison of the greedy algorithm with more sophisticated techniques offers a lot of value to the paper.

**Other Comments Or Suggestions:**

- In section 1.1: with multiple attributed data, not multiply
- In section 5.1: considered both regular and skewed graphs, not consider
- One suggestion is that it is weird that the first figure mentioned is 2(b) instead of 2(a), is there a reason these figures are not flipped?

**Other Strengths And Weaknesses:**

One main strength of the paper is its originality and significance. I think this problem has a lot of important applications and this sheds a lot light on the important techniques that can be used.
The paper is generally also well written, although it has some typos which I list in the section below.

**Questions For Authors:**

I first have the two questions from the claims section, which I copy here:
- The claim that allowing duplicates allows for a larger dataset which reduces DP noise. I fail to understand why having duplicates leads to a larger dataset, as we would still have the same limit on contribution.
- In section 5.2, the authors hypothesize that the skew in the graph is because k more users will have <k examples than for the regular graph and then the dataset will be smaller. Again, I fail to see why this is true.

Also, I have some other questions:
- In section 2, the definition of the attributed dataset, this seems like $x_i$ can not be repeated in $e_i$? Can you make this clearer whether it can?
- Also same section when defining edge data DP, it is said that m=m' and $e_i=e'_i$ for all $i\in [m]$ but differ in one $x_i$. But doesn't some $e_i$ change when changing $x_i$?

**Relation To Broader Scientific Literature:**

The problem at hand is a very important advancement in the literature as it discusses the important and natural extension of multi-attribution. In the real world, in many cases, such as the emails example in the paper, multiple users contributing to a single example needs to be taken into account along with the user-level privacy where a user contributes to multiple examples.

**Theoretical Claims:**

There are not really theoretical proofs in the paper as the paper empirically justifies and tests the proposed methods, as well as compares the different techniques.

---

> ### Author Rebuttal · Authors · 2025-03-31
>
> Thanks for your support of the paper, and for the editing suggestions which we plan to incorporate. Here we respond to some of the reviewer’s questions. As space permits, we will plan to include more detailed explanations along these lines in the revision:
> * __“X_i cannot be repeated in e_i”__: We believe this question is asking whether two different edges can have examples with the same content. Our definition does not restrict the values of different x_i, even in relation to each other. This is in line with other DP definitions such as example-level DP or user-level DP in the single attribution model, which do not make any assumptions on the contents of examples in relation to each other.
> * __“Doesn’t some e_i change when changing x_i”__: Here we are decoupling the edge (i.e. users associated with an example) and the content of the example. e.g. For edge-data adjacent datasets D, D’, in D Alice might send Bob the email “how was your surgery?” and in D’ Alice might send the email “are you free for lunch tomorrow?”. In both cases e_i = {Alice, Bob}, but the content x_i has changed (and for edge-data adjacency, the rest of the graph is unchanged).
> * __“duplicates allows for a larger dataset which reduces DP noise…”__: We apologize for not elaborating further in the paper. As a simple example, suppose our users are {A, B, C, D} and our examples are associated with hyperedges {{A}, {A, B}, {A, C}, {B, C}, {A, D}}. If we use k = 3, without duplicates we can get a dataset of size at most 4, since we have to discard one of the examples including A. However, with duplicates we can use the dataset, e.g. {{A}, {A, B}, {A, D}, {B, C}, {B, C}} which includes 5 examples and still satisfies the bound k = 3. That is, we can use duplication (equivalently, upweighting of some examples) to even out users’ contributions and make full use of each user’s allocation.
> * __“more users will have <k examples than for the regular graph and then the dataset will be smaller.”__: For simplicity let’s focus on 2-uniform graphs, i.e. the usual graphs where every edge has two users. Consider the 2-regular graph {{A, B}, {B, C}, {C, D}, {D, E}, {E, F}}. For contribution bound k = 2, by including all edges, every user saturates their contribution bound and we get 6 edges. In contrast, consider the graph {{A, B}, {B, C}, {C, A}, {A, D}, {B, E}, {C, F}}. The number of users/edges is the same, and the average degree is still 2, but half the users have degree 1 and half have degree 3. So under a contribution bound of 2, the best we can do is to include 4 edges (e.g. {{A, B}, {A, D}, {B, E}, {C, F}}. In short, some of the users don’t have enough edges to saturate their contribution bound, and others have too many edges and must discard some, so we can include fewer edges than if the graph were perfectly regular.

---

### Official Review · Reviewer_Eu7P · 2025-03-14

**Overall Recommendation:** 4

**Summary:**

The paper studies user-level differential privacy when each training sample can be attributed to multiple users, called multi-attribution model. It proposes a new privacy definition called fixed-graph DP,  where users are nodes and examples are hyperedges. And the neighboring database is defined as arbitrary changes to all examples associated with a single user, which is similar to node-dp in graph dp.

To apply DP-SGD, DP-MF type of algorithms, we need to select a subset of the dataset where each user contributes a limited number of examples. This is proposed as contribution bounding problem, which is NP-hard. The authors present a greedy algorithm for contribution bounding and empirically evaluate it. The paper compares DP-SGD and DP-MF, finding DP-SGD generally outperforms DP-MF in the multi-attribution setting due to privacy amplification challenges with DP-MF.

**Claims And Evidence:**

I think there are two claims: 1) The authors proposed a new setting called this multi-attribution model. 2) the proposed greedy algorithm could be used for practical training. The first claim is interesting and practical. The second claim is evaluated based on experiments.

**Essential References Not Discussed:**

None

**Experimental Designs Or Analyses:**

Yes. There are two tasks: training a small transformer on the arXiv dataset, and a synthetic logistic regression task. Both are relatively small scale, but I think it is convincing.

**Methods And Evaluation Criteria:**

Yes.

**Other Comments Or Suggestions:**

In those figures, I feel like epsilon larger than 10 does not make too much sense. It is nearly non-private.

**Other Strengths And Weaknesses:**

Strengths: presentation is clear. Problem is interesting.
Weaknesses: the greedy algorithms are somewhat simple and lack of theoretical novelty. I guess this problem would encourage more research in the future.

**Questions For Authors:**

I feel like favoring examples with fewer users could skew the model toward less active or less connected users, potentially harming algorithmic fairness. Do you have any insights or any thoughts on this?

**Relation To Broader Scientific Literature:**

This new problem setting multi-attribution model could potentially bring more study and research.

**Theoretical Claims:**

No theoretical proof.

---

> ### Author Rebuttal · Authors · 2025-03-31
>
> Thanks for your support of the paper. We respond to a few points below:
> * __"The greedy algorithms are somewhat simple.”__: We agree the algorithms are simple, and we have framed them as baselines to emphasize this. We note that our empirical results, which demonstrate the baseline algorithms are quite competitive, suggest developing much more complicated algorithms may be unnecessary in practice (but may still be an interesting research direction). We also agree that part of the goal of this paper is to encourage more work on the multi-attribution setting.
> * __“epsilon larger than 10 does not make too much sense. It is nearly non-private.”__: We agree that in practice epsilons larger than 10 may not give meaningful privacy guarantees. Our goal with studying large epsilons was mainly to extend our empirical understanding to low-noise regimes. We also note that e.g. an epsilon of 64 in one setting can correspond to a much smaller epsilon in a different setting with the same noise multiplier and batch size but e.g. a different dataset size or graph structure, so studying these low-noise regimes may provide useful insights even if one does not believe that the corresponding epsilon is meaningful in practice.
> * __“Potentially harming algorithmic fairness”__: This is an interesting point. Note that contribution bounding also makes the contributions of different users more uniform, so even without differential privacy one might expect contribution bounding to actually increase fairness in this dimension. One could summarize the impact on fairness as, contribution bounding favors examples with fewer users and users with fewer examples. We expect that whether this leads to an overall positive or negative impact on fairness will be highly subjective and context-dependent, depending on what properties of the dataset these quantities (examples per user and users per example) align with.

---

### Official Review · Reviewer_gGTR · 2025-03-14

**Overall Recommendation:** 4

**Summary:**

The paper introduces a novel differential privacy (DP) definition for datasets with multi-user attribution, where each training example is associated with multiple users (e.g. emails attributed to both senders and recipients).

Their proposed adjacency definition, termed "fixed-graph DP," protects the content of each edge (message) but not the graph structure. This approach enables the design of significantly more practical algorithms compared to the more conservative existing definition (Node DP, Fang et al., 2022).

The authors address the challenge of bounding user contributions through example selection (an NP-hard problem). They first propose a greedy algorithm and then attempt to improve it using linear programming techniques, though with limited practical impact.

The paper includes comprehensive experiments that demonstrate optimal strategies for both data selection and DP training algorithms, exploring the tradeoffs between bias and variance in the multi-attribution setting.

## Post-rebuttal update

I choose to maintain my high score. Although I agree that lack of formal protection of a graph structure is concerning, this paper is a first of its kind and proposes a new DP formulation - paving the way for the future work, which can address the issue.

**Claims And Evidence:**

The paper presents a comprehensive body of work - they introduce a new, well-justified definition of privacy, develop methods to bound user contributions, and effectively apply existing privacy accounting techniques. All aspects of their approach are thoroughly proven as necessary.

The authors explore a good range of algorithmic design options to provide fixed-graph DP. Their investigation covers greedy algorithms, linear programming techniques, and variants that handle duplication and bias mitigation.

The experimental results effectively demonstrate the viability of their approach across different settings, showing performance on both synthetic and real-world datasets.

Overall, I believe this is a very strong paper with a novel and highly relevant idea, backed by proper evidence and rigorous analysis.

However, I note two significant gaps:

1. The paper argues that the additional public data (the graph structure) poses minimal risk because it would be difficult for a practical attacker to exploit given the algorithm design. However, graph structure can often be highly sensitive information (including in their email example). More evidence is needed to support their claim that this doesn't undermine the privacy guarantees - otherwise, the provided guarantees may not be as meaningful as presented.

2. The paper compares their approach to a stronger definition (Node DP), arguing that their definition relaxation allows for more practical algorithms. I would like to see a comparison on "nice" graphs - cases where Node DP has practical implementations - to better understand at what point fixed-graph DP becomes an essential tradeoff. This would help clarify when each approach is most appropriate.

**Essential References Not Discussed:**

None

**Experimental Designs Or Analyses:**

See "Methods And Evaluation Criteria"

**Methods And Evaluation Criteria:**

The paper proposes a well-justified set of methods to address privacy in the multi-attribution setting.

The experimental methodology is sound, utilizing both synthetic data (allowing for controlled investigation of specific parameters) and real-world arXiv data (demonstrating practical applicability). The authors thoughtfully explore key algorithmic tradeoffs, particularly the balance between noise reduction and bias mitigation across different privacy budgets.

As noted previously, however, a more extensive comparison with node DP would significantly enhance the paper. While the authors convincingly argue for the theoretical advantages of their approach, empirical comparisons on graphs where node DP solutions are feasible would provide better context for understanding when fixed-graph DP becomes an essential tradeoff.

**Other Comments Or Suggestions:**

N/A

**Other Strengths And Weaknesses:**

N/A

**Questions For Authors:**

N/A

**Relation To Broader Scientific Literature:**

The paper addresses a highly relevant question of providing DP guarantees on data simultaneously belonging to multiple users. Despite being common in many realistic and sensitive datasets (emails, messages, collaborative documents), the DP literature has lagged in providing appropriate guarantees for such scenarios.

While user-level DP has been extensively studied, its application to multi-attribution settings remained largely unexplored. Fang et al. (2022) proposed Node DP to address this gap, but it had significant practical limitations. This paper makes a valuable contribution by introducing fixed-graph DP as a more practical alternative, bridging an important gap in the privacy literature.

**Theoretical Claims:**

The paper introduces a new privacy definition (fixed-graph DP) that's well-formulated for the multi-attribution setting. The authors rigorously prove that their contribution bounding algorithms satisfy this definition and demonstrate how existing DP-SGD and DP-MF accounting techniques can be appropriately applied in this context.

The theoretical analysis is sound, properly establishing the properties of their algorithms while acknowledging computational limitations like the NP-hardness of the optimal solution.

Overall, the theoretical foundation is solid and well-substantiated.

---

> ### Author Rebuttal · Authors · 2025-03-31
>
> Thanks to the reviewer for their support of the paper. We want to respond to the two gaps raised by the reviewer.
>
> * Gap 1: We agree that the graph structure can be a privacy risk. We do not advocate that _every_ algorithm that achieves fixed-graph-DP is appropriate for ML settings, and do not advocate for actually publishing the graph in practice. While we do not formally protect the graph structure in a DP manner, as discussed at the end of Section 2, the structure of our algorithms (which discard the graph information during training) means they resist straightforward attacks (with non-pathological data). Part of our goal with this work is to invite future research on this problem, including designing attacks on our two-phase algorithms in practical settings.
> * Gap 2: We agree such a comparison could be useful. However, to the best of our knowledge, the only setting where batch gradient queries with node DP would be readily applicable and feasible at scale is when the hypergraph is already bounded-degree, in which case contribution bounding is a no-op and fixed-graph DP/the comparison are unnecessary. While reductions (by truncating some nodes) to bounded degree graphs are known, these reductions are only understood for “2-uniform” graphs (that is, the usual graphs, where edges involve only two nodes, as opposed to hypergraphs, which are the focus of our work).

---

### Official Review · Reviewer_z4vo · 2025-03-15

**Overall Recommendation:** 2

**Summary:**

The work describes how to apply two differentially private training methods for data in which more
than one individual can contribute to each data instance (i.e. email sender and receivers, or
author set in a publication). The approach is based on building a subset of the dataset so as to
upper bound the number of contributions (i.e. emails involved in, papers authored) each individual
is involved in (as a configurable parameter). Several mostly greedy methods are used to find this
subset. Once this subset is available, existing DP training methods can be used. The subsetting
method does not incur any privacy cost due to the paper's interpretation of an individuals'
contribution to a dataset: their relationships to instances are not private, only the instance
contents are (i.e. the contents of emails or papers are considered private but authorship or
sender/receivers of emails are not).

Experiments exemplify the method on arXiv paper abstracts with contribution defined by authorship.
To investigate some aspects of global authorship metrics, synthetic datasets where authorship is
generated are also analyzed. First, the greedy subsetting methods are demonstrated to be relatively
close to the optimal by comparing them to solutions derived using integer programming. For the
experimental task, masked token prediction is used. A small BERT model is fine-tuned using two DP
SGD methods with data drawn as per the proposed methodologies. A baseline without DP is also
included. The experiments show mostly expected results with respect to the privacy parameter though
also shown is that both DP SGD methods have situations under which they perform better.

## update after rebuttal

Discussion regarding edge/node privacy suggested there is a path towards acceptance if the paper
better presents its limitations and consequences of those limitations both in terms of privacy and
what sorts of ML methods the formalism can be applied to (i.e. not graph ML). I have raised my
score to weak reject. The bulk of my other objections, the levels of novelty and contributions,
will need to be evaluated by the AC for the final decision.

**Claims And Evidence:**

Yes. Note that the paper seems to avoid claiming anything. Some speculation is done around the
experimental results but nothing one could object to. This is a weakness which I discuss further in
the weaknesses section.

**Essential References Not Discussed:**

I did not check. The paper has sufficient weaknesses already.

**Experimental Designs Or Analyses:**

Yes.

**Methods And Evaluation Criteria:**

Yes.

**Other Comments Or Suggestions:**

See Weaknesses section above for suggestions.

Comments

- C1. It is difficult to connect how the synthetic dataset construction models aspects of the real
  arXiv dataset. One thing that can help is to report quantitative metrics about both synthetic and
  real, answering the question: the synthetic dataset with parameter choice $ p $ causes metric $ m
  $ a function of $ p $ and on the real dataset, this metric is equal to $ m' $.

- C2. Considering repeating experiments that feature non-determinism (even in the train split
  picking) and demonstrate results alongside margins (under some chosen confidence). Many results
  as plotted seem very close to each other, making it difficult to derive conclusions without
  considerations of statistical confidence.

Small things:

-

Smallest things:

- Several uses of "natural" in the first 2 paragraphs of Section 4 could be reworded to explain the
  naturalness instead of assuming it is obvious.

**Other Strengths And Weaknesses:**

Strengths:

+ S1. Very well written introductory and background sections.

Weaknesses:

- W1. No take-aways or claims that can inform application of the methodology.

  - Examples of suboptimality of the greedy methods shows trivial reduction of number of selected
    nodes as compared to the optimal method but no claim is made about any suboptimality bounds. It
    is not clear to what degree one can generalize the experimental observations.

  - Impact of subsetting on masked model difficult to assess. No baseline is shown in Figure 2 to
    demonstrate how much utility is lost due to either the proposed method or DP-SGD/MF.

  - Suggestion G1. Perform sufficient experimentation to write some take-aways that generalize
    them. If possible, prove conditions under which suboptimality can be bounded. Add baselines to
    all experiments which should also help with take-aways. Include a Conclusions sections that
    summarizes the main take-aways.

- W2. Objectionable assumption about privacy in the setting. The "Fixed-graph (multi-attribution
  user-level) DP" seems to make the assumption that attribution graph itself is not private. A
  mechanism that outputs the graph adjacency in the plain (without the hyper-edge data xᵢ) would be
  ε=0 DP. While user identities (i.e. email addresses) are not explicitly modeled, the disregard of
  adjacency in the privacy calculus suggests that email sender/recipients or authorship are
  not private/sensitive information.

  The argument for why a recipient of an email should be considered a contributor is that the email
  might be "about" them. A fact that someone is a recipient may have as much if not more
  information about them than the content of the message.

  Database pairs D, D' as per Definition 2.1 differ only in message/abstract contents. This is not
  the difference one would expect due to the choice of an individual to participate in the dataset
  or not. Under what circumstances is a user deciding or not to participate in an email/abstract
  dataset but is not making the choice of being included or not, but instead is making the choice
  of the email contents/abstracts being included or not? Including empty emails/abstracts for an
  emails sent to/received by an individual who wishes not to participate would already be a
  violation of their agency to consent.

  - Suggestion G2. Either provide scenarios under which the privacy assumptions make sense (email
    and authorship are not viable as per above discussion) or switch to the Node privacy setting.

- W3. Limited novelty. The novelty in the "multi-attribution" setting (each instance may have had
  contributions from more than 1 individual) doesn't seem to matter to the methodology. The same
  identical arguments and method could have been used in the group privacy setting without
  multi-attribution. The impact of multi-attribution is on how large the subgraphs can be as per
  produced by the methods but those greedy methods do not seem to be tailored in any way to
  multi-attribution.

  - Suggestion G3. One option is to significantly expand on the take-aways to demonstrate how
    viable subgraphs with k-bounded attribution are for DP-SGD/MF under a wide range of datasets,
    tasks, models, etc. This would be more or less a paper focused on addressing W1 above.
    Alternatively, alternate approaches or techniques that are specific to multi-attribution can be
    presented and experimented with.

**Questions For Authors:**

- Q1. Did the BERT-Tiny model already have masked token prediction layers pre-trained? If so, how
  did it perform the experimental task before fine-tuning on the arXiv abstracts?

**Relation To Broader Scientific Literature:**

The methodology's privacy arguments reduce to DP group privacy. Methods (graph building with
attribution bounds) are simple enough that they are not based on any recent work. Experimentation
uses DP-SGD and DP-MF training methods from private ML research.

**Theoretical Claims:**

The paper makes no theoretical claims. There is a point about NP-hardness in the appendix but it
is mostly a citation.

---

> ### Author Rebuttal · Authors · 2025-03-31
>
> Thanks to the reviewer for their feedback. We agree that broadly, there is much more work to be done to fully understand the multi-attribution model, and part of our goal with this work is to motivate future research into this setting. Below we respond to individual points we disagree with in the review:
> * W1:
>   * __“no claim is made about any suboptimality bounds”__: As mentioned in Appendix B.2, theoretically the problem is known to be hard to approximate to within large factors, even in some very simple settings. So while one could attempt to prove positive theoretical results, these would be overly pessimistic and not give meaningful guidance in practice.
>   * __Q1/“add baselines”__: We use the checkpoint available [here](https://github.com/google-research/bert) as our initialization, which achieves a test loss of 4.955 for arXiv, i.e. is not competitive with DP fine-tuning even at small epsilons. We will state this as a reference point for the empirical results in the revision. Since we introduce fixed-graph DP (and as we discuss in our response to reviewer gGTR, the node-graph DP algorithms are not readily applicable except in trivial settings), besides the not-fine-tuned checkpoint and some other baselines we already included in the paper, we do not feel there are obvious and meaningful baselines to compare our greedy baseline to. However, we are open to suggestions from the reviewer for other baselines to consider adding in the revision.
> * W2:
>   * We are glad the reviewer raised the question of appropriate DP formulations; in fact, one of the main goals for our paper is to provide a better framework for such discussions. Node-DP is not well known in the ML community currently, and hopefully our paper will provide a clear critique to simply applying example-level or user-level DP to settings where examples contain information from multiple users.
>   * At the same time, we also agree that the fixed-graph DP is not as strong as one ideally would want. We spent considerable time looking for ways to scale node-DP, and nevertheless this approach seems very far from being feasible for large-scale ML applications. Given the explosion of GenAI, there is a pressing need for practical approaches in this space that can be offered now.
>   * We do not advocate that _every_ algorithm that achieves fixed-graph-DP is appropriate for ML settings. Our algorithm carefully decouples the use of the graph from the use of the training data. This decoupling excludes the most obvious problematic algorithms (i.e., ones that publish the graph). In fact, despite considerable effort, we were unable to design any realistic attack to exploit our algorithms. At the same time, we also lack a fully satisfactory guarantee. (For example, one can prove statements assuming independence of the training data from the graph structure; but such independence does not hold in practice and we find it unsatisfactory.)
>   * Hence, we propose an approach here that is practical and provides significantly better privacy than the naive application of existing DP ML solutions. Equally importantly, we hope our work highlights the problem, and stimulates future research. We do not advocate for our approach as a complete solution, but feel it is a valuable starting point that already highlights nontrivial choices and phenomena.
> * W3:
>   * __On lacking novelty__: While our greedy algorithms are not very complicated, we believe the fixed-graph DP definition and the contribution bounding framework are meaningful contributions, and we believe our empirical results developing a better understanding of the greedy baseline raise and study non-trivial questions about contribution bounding.
>   * __On tailoring to multi-attribution__: While the greedy algorithm retrieves the trivial strategy maximizing |S| in the single-attribution case, most of our empirical results answer questions which do not make sense to ask in the single-attribution case (e.g. in the single-attribution case, there is no difference from the ILP, no bias towards examples with fewer users). Furthermore, designing a more complicated method tailored to the multi-attribution case is unnecessary if the greedy algorithm is competitive with strong baselines like the ILP solver, which our empirical results provide evidence for.

---

### Decision · Program_Chairs · 2025-05-01

**Decision:**

Accept (poster)

**Comment:**

The paper is about differential privacy in a setting where each piece of data is possibly attributed to multiple users. It argues that existing DP models for this setting do not scale to modern-ML level, and puts forward an alternative definition called fixed-graph DP. It shows that simple methods to select a subset of the data with bounded attributions (which facilitate DP) perform well empirically.


Reviewer discussion was mostly in consensus about content and substance, despite the wide range of scores. The strength of the paper is in addressing the important matter of new privacy definitions that strike a balance between privacy and feasibility in modern ML. Even if the specific definition given in the paper is not without its flaws, reviewers are hopeful it would spur more research and discussions in this direction, and this appears to be the authors' mindset as well. The weaknesses discussed were about the limited technical contribution of the paper - at the technical level, it only proposes a definition and tests some methods which the authors willingly concede are simple baselines (the contribution is in establishing that even simple baselines already perform well empirically).


Another particular weakness discussed was the role the graph structure plays in the model, a point that the paper is somewhat vague about. It appears the paper's definition does not consider the graph structure public, yet does not formally protect it. Rather, the authors' "two-phase" methods remove the graph structure from the input to the downstream algorithm, which on the one hand limits applications since it precludes algorithms that rely on the explicit graph structure, but on the other hand does not formally protect the graph structure which could still leak through correlations with user data. Reviewers advise the authors to expand and clarify the discussion of this point in the final version of the paper, and to add a discussion of limitations this might create in applications to graph ML methods.


In the end, the overall opinion was positive about accepting this paper, ideally under a revision that addresses the points above.